# Phylogenetic Implication of Large Intergenic Spacers: Insights from a Mitogenomic Comparison of *Prosopocoilus* Stag Beetles (Coleoptera: Lucanidae)

**DOI:** 10.3390/ani12131595

**Published:** 2022-06-21

**Authors:** Mengqiong Xu, Shiju Zhou, Xia Wan

**Affiliations:** 1Department of Ecology, School of Resources and Environmental Engineering, Anhui University, Hefei 230601, China; xumengqiong1226@126.com (M.X.); 15056161961@126.com (S.Z.); 2Anhui Province Key Laboratory of Wetland Ecosystem Protection and Restoration, Hefei 230601, China

**Keywords:** lucanid beetles, mitogenomes, phylogeny

## Abstract

**Simple Summary:**

Insect mitochondrial genomes (mitogenomes) show high diversity in some lineages. In the mitogenome of some Coleoptera species, a large intergenic spacer (IGS) has been identified. However, very little is known about mitogenomes of lucanid beetles. In this work, to provide further insight into the phylogenic relationships among species in lucanid beetles (genus *Prosopocoilus*), two *Prosopocoilus* species (*Prosopocoilus castaneus* and *Prosopocoilus laterotarsus*) were newly sequenced and comparatively analyzed. Significantly, the two newly sequenced *Prosopocoilus* species contained a large IGS located between *trnI* and *trnQ*. Our phylogenomic analyses showed that *P. castaneus* and *P. laterotarsus* were clustered in a clade with typical *Prosopocoilus* species (*Prosopocoilus confucius, Prosopocoilus blanchardi,* and *Prosopocoilus*
*astacoides*). These results provide valuable data for the future study of the phylogenetic relationships in this genus.

**Abstract:**

To explore the characteristics of mitogenomes and discuss the phylogenetic relationships within the genus *Prosopocoilus*, the mitogenomes of two species (*P. castaneus* and *P. laterotarsus*) were newly sequenced and comparatively analyzed. The arrangement of the mitogenome in these two lucanid beetles was the same as that in the inferred ancestral insect, and the nucleotide composition was highly biased towards A + T as in other lucanids. The evolutionary rates of 13 protein-coding genes (PCGs) suggested that their evolution was based on purifying selection. Notably, we found evidence of the presence of a large IGS between *trnI* and *trnQ* genes, whose length varied from 375 bp (in *P. castaneus*) to 158 bp (in *P. laterotarsus*). Within the large IGS region, a short sequence (TAAAA) was found to be unique among these two species, providing insights into phylogenomic reconstruction. Phylogenetic analyses were performed using the maximum likelihood (IQ-TREE) and Bayesian (PhyloBayes) methods based on 13 protein-coding genes (PCGs) in nucleotides and amino acids (AA) from published mitogenomes (*n* = 29). The genus *Prosopocoilus* was found to constitute a distinct clade with high nodal support. Overall, our findings suggested that analysis of the characteristics of the large IGS (presence or absence, size, and location) in mitogenomes of the genus *Prosopocoilus* may be informative for the phylogenetic and taxonomic analyses and for evaluation of the genus *Prosopocoilus*, despite the dense sampling materials needed.

## 1. Introduction

Mitogenomes have been widely applied to analyze population genetics, phylogenetics, and molecular evolution across insect species [1,2,3]. Mitogenomes could be reliable, convenient, and useful markers owing to their simple genomic structure, a faster rate of evolution, and maternal inheritance patterns across different groups of insects [3,4,5,6,7,8,9,10]. More recent studies in the Coleoptera families, Scarabaeidae [4], Chrysomelidae [11], Curculionidae [12], and Meloidae [8,13] have shown that the characteristics of mitochondrial genomes, such as genome organization, base composition, protein-coding genes, codon usage, intergenic spacers, control regions, etc., could be informative for phylogenetic analyses. Intergenic spacers (IGSs) are non-coding regions typically found in metazoan mitochondrial genomes [14]. For example, in some species of beetle, known mitogenomes contain the IGS commonly found at the junction of the *trnS2 (UCN)* and *nad1* genes [8,15,16,17,18,19,20]. The IGSs have also been found between different genes in the mitogenomes of other beetles, such as between *trnW* and *trnC* [8], *nad2,* and *trnW* [21], *trnI* and *trnQ* [4,22,23], and so on. Comparative analyses of IGSs suggest that the presence and location of the IGS may be useful for studying evolutionary history and species delimitation in insects [24,25], such as those in the families Meloidae [8,13], Cleridae [16], and Scarabaeoidae [18]. Although a recent study reported that the *Dorcus velutinus* complex had a large IGS (>50 bp) between *trnS2 (UCN)* and *nad1* [20], very little is known about large IGSs in *Prosopocoilus* mitogenomes.

The genus *Prosopocoilus* Hope and Westwood (1845) contains approximately 200 species and subspecies, accounting for approximately 15% of all known stag beetles in the world, with distributions across the African, Australian, Oriental, and Palearctic realms [26,27]. Like most stag beetles, members of *Prosopocoilus* show strong sexual dimorphism and male polymorphism; there are distinctly conspecific variations in shape and size of the mandible and head and sometimes in body colour. However, many species are so highly divergent in their morphological traits that it is difficult to classify them as *Prosopocoilus*. Consequently, many taxonomic changes and adjustments have been made to establish more “reasonable” positions for these taxa due to different opinions held by different taxonomists [26,28,29,30,31,32,33,34]. Systematics of *Prosopocoilus* has, to date, been a problematic issue in the Lucanidae. Accordingly, there is an urgent need for additional molecular phylogenetic studies. Mitogenomic datasets are powerful molecular tools for exploring the evolution of Coleoptera [6,16]. Unfortunately, there have been very few gene fragments and only three identified mitogenomes to date [1,4,23,35]. Of the approximately 200 species within *Prosopocoilus*, only three mitogenomes (those of *Prosopocoilus confucius* Hope, *P**. blanchardi* Parry, and *P**. astacoides* Hope) have been described [4,23]. These studies of *Prosopocoilus* mitogenomes demonstrated that both species maintained the ancestral type for insects without rearrangement, but possessed a large non-coding region, excluding the control region [4,23]. The lack of research greatly limits the mitogenomic comparisons and molecular phylogenomic studies of the genus *Prosopocoilus*. Thus, we considered it a priority to explore the mitogenome evolution in the genus *Prosopocoilus*.

Accordingly, in this study, we newly sequenced and annotated the complete or almost complete mitochondrial genomes of two species belonging to *Prosopocoilus* (*P. castaneus* Hope & Westwood and *P. laterotarsus* Houlbert). We described and compared the characteristics of these two newly sequenced mitogenomic sequences, together with 24 previously sequenced lucanid mitogenomes. Additionally, we analyzed the large IGS in the mitogenomes of all studied *Prosopocoilus* species to explore their evolutionary significance. The large IGS may provide phylogenetic signals. Most notably, for the five species sequenced in the present study, the large IGS is located between *trnI* and *trnQ*, including a five bp long short sequence (TAAAA) with different repeat patterns within this uniquely large IGS region. Additionally, we also find this distinctive feature of the large IGS region to be useful for a systematic understanding of the genus *Prosopocoilus*.

## 2. Materials and Methods

### 2.1. Sample Collection and DNA Extraction

Data were collected for the following two newly sequenced specimens: male *P. castaneus*, Lincang, Yunnan, China, July 2017; and female *P. laterotarsus*, Medog, Xizang, China, June 2016. The voucher specimens were deposited in the Museum of Anhui University, China. At the time of sampling, muscle tissues were placed in pure ethanol. Total genomic DNA was extracted from the thorax using a Blood and Tissue Kit (Qiagen, Germany). The newly generated mitogenomic sequence data have been deposited into the GenBank database with the following accession numbers: ON401054 for *P. castaneus* and ON401055 for *P. laterotarsus*.

### 2.2. Polymerase Chain Reaction (PCR) Amplification and Sequencing

PCR amplification reactions for *cox1*, *cytb*, and *16S* were performed in a final volume of 25 μL, including 8.5 μL sterile double-distilled water, 12.5 μL 2 × EasyTaqSuperMix (+dye), 2 μL template DNA, and 1 μM of each primer (forward and reverse). Three amplification reactions were used as “anchors” to assemble complete mitogenomes. All primers used for PCR amplification are summarized in Table 1. The PCR amplification reaction conditions were the same as those described by Lin et al. [4]. For mitogenome sequencing, we prepared a library using a Truseq nano DNA kit (Illumina) with an insert size of 450 bp and sequenced it on the Illumina HiSeq 2000 platform at Berry Genomics (Beijing, China). Raw reads were trimmed of adapters using Trimmomatic [36], and low-quality and short reads were removed with Prinseq [37].

### 2.3. Mitogenome Assembly, Annotation, and Analysis

High-quality reads were applied to de novo assemble using IDBA-UD [38] with minimum and maximum k values of 80 and 240 bp, respectively. We then evaluated the accuracy of the assembly by re-mapping clean reads to the mitogenome assemblies in Geneious v6.1.7 (Biomatters Ltd., Auckland, New Zealand), allowing mismatches of up to 2%, a minimum overlap of 100 bp, and a maximum gap size of 3 bp. The preliminary annotations were made on the MITOS Web Server with the invertebrate mitochondrial code (http://mitos.bioinf.unileipzig.de/inxdex.py, accessed on 12 September 2021). Protein-coding regions were identified by aligning published mitochondrial sequences in Geneious v6.1.7 (Biomatters Ltd.), whereas tRNA genes and their secondary structures were inferred using tRNA scan-SE v2.0 [39]. Owing to they could not be identified using tRNA scan-SE, 16S ribosomal RNA (*rrnL*) and 12S ribosomal RNA (*rrnS*) were determined according to sequence similarity with related species. Relative synonymous codon usage (RSCU) and Nucleotide composition were computed using MEGA-X [40]. Composition skew analysis was performed using the following formulas: AT skew = (A – T)/(A + T) and GC skew = (G – C)/(G + C) [41]. The evolutionary rates (Ka/Ks ratios) for each PCG were calculated using DnaSP v5.0 [42]. The mitogenomic map was drawn using the CG View server V 1.0 [43].

### 2.4. Phylogenetic Analyses

Phylogenetic analyses were performed using five *Prosopocoilus* stag beetles, twenty-one other stag beetles as an ingroup, and three scarab beetles (*Cheirotonus jansoni*, *Protaetia brevitarsis,* and *Rhopaea magnicornis*) as an outgroup (Table 2). Protein coding genes (PCGs) of these species were extracted based on GenBank annotations using GenScalpel [44]. The nucleotide and amino acid sequences of the 13 PCGs were used as the dataset to construct the Bayesian inference (BI) and maximum likelihood (ML) phylogenetic trees, respectively. All these nucleotide and amino acid sequences were aligned using translation align and geneious align, respectively, in the programme Geneious 9.0.5. Gaps and ambiguous sites were removed from the protein alignment to generate a 10,878 bp nucleotide dataset and a corresponding amino acid dataset (3626 amino acids). We then set the model selections as Akaike information criterion (AIC), greedy search algorithm, and unlinked branch lengths to estimate the best fitting schemes in the programme PartitionFinder 2.1.1 [45]. The best-fit partitioning schemes and evolutionary models for the nucleotide and amino acid datasets are presented for ML analyses (Appendix A).

BI and ML analyses were conducted using PhyloBayes MPI 1.5a [46] and IQ-TREE web server [47], respectively. Then, BI analysis was performed on the CIPRES Science Gateway [48] and the site-heterogeneous mixture model (GTR + CAT) was chosen [49]. Two independent chains starting from a random tree were run for 20,000 cycles, with trees being sampled every 10 cycles. The initial 25% of trees for each MCMC run were discarded as burn-in. A consensus tree was computed from the remaining 1500 trees combined from two runs, and the two runs converged at a maxdiff of less than 0.1. In ML analyses, the “Auto” option was set under optimal evolutionary models, and the phylogenomic trees were constructed using an ultrafast bootstrap approximation approach with 10,000 replicates. Phylogenomic trees were viewed and rooted with the three species in Scarabaeidae as the outgroup in Figtree v1.4.3 (http://beast.bio.ed.ac.uk/figtree, accessed on 1 June 2022).

## 3. Results and Discussion

### 3.1. Genome Organization and Base Composition

In this study, we obtained one complete mitogenome of *P. castaneus* and one nearly complete mitogenome of *P. laterotarsus*. The total lengths of the two mitogenomes ranged from 17,523 bp (*P. castaneus*) to 17,333 bp (*P. laterotarsus*). Both new mitogenomes shared the ancestral type for insects [55,56]; thus, these mitogenomes were composed of 13 PCGs, 22 tRNAs, two rRNAs, and a control region. We recovered only a partial control region of the mitogenome of *P. laterotarsus* (Table 3). Of the 37 genes, nine PCGs and 14 tRNA genes were located on the majority strand (J-strand), with the remaining four PCGs, two rRNA genes, and eight tRNAs genes on the minority strand (N-strand) (Figure 1). The mitochondrial genome map of *Prosopocoilus* species is shown in Figure 1. Both mitochondrial genomic arrangements were similar to other published stag beetle mitogenomes [4,20,23,50]. All 22 tRNA genes were detected in these two newly sequenced mitogenomes (Table 3, Figure 1). The lengths of the tRNA genes ranged from 60 to 70 bp (Table 3). All tRNA genes displayed the representative clover-leaf secondary structure (Appendix A), whereas *trnS1 (AGN)* lacked the dihydrouridine (DHU) arm, and it was replaced with a simple loop, which was common in other lucanid mitogenomes [4,23,50,51,52,53,54]. There is a similar nucleotide proportion among the two newly sequenced mitogenomes, with a high content of AT (Table 4). The AT skews of PCGs, tRNA genes, and rRNA genes in the *P. castaneus* and *P. laterotarsus* mitogenomes were −0.16/−0.18, 0.05/0.05 and −0.11/−0.08, respectively (Table 4). The AT-skew values were negative in PCGs but positive in the tRNA genes within the two mitogenomes, consistent with those found in other lucanid mitogenomes [4,20,23,50]. Additionally, the nucleotide compositions of lucanid mitochondrial genomes corresponded well to the AT skew generally observed in other beetles [8,11,12,13].

### 3.2. Protein-Coding Genes and Codon Usage

The lengths, nucleotide proportion, and codon usages of 13 PCGs in these two new mitogenomes were nearly the same as those in the ancestral type of insects (Table 3 and Table 4; Figure 1 and Figure 2). All 13 PCGs were identified in these two new mitogenomes. Twelve PCGs of these two new mitogenomes used ATN (where N represents A, C, G, or T) as initiation codons, with the exception of *cox1*, which was initiated with AAN. Notably, AAN is an accepted conventional start codon in many beetle mitogenomes [8,11,15,23,57]. In the *P. castaneus* and *P. laterotarsus* mitogenomes, six PCGs shared the typical stop codons TAA and TAG, whereas in the remaining genes, an incomplete stop codon consisting of T or TA was inferred (Table 3). All of the new mitogenomes had incomplete stop codons, as described in other stag beetles [4,23,51] and other insects [58,59], which have been demonstrated to produce functional stop codons in polycistronic transcription cleavage and polyadenylation processes [8,13,14]. Comparisons of five mitogenomes of *Prosopocoilus* stag beetles showed that the *cox1* genes of *P. castaneus*, *P. laterotarsus*, *P. blanchardi*, *P. astacoides*, and *P. confucius* shared the same incomplete stop codon (T). By contrast, complete terminators (TAA) are utilized in *Kirchnerius guangxii*, *Epidorcus gracilis,* and the related species *Serrognathus platymelus*. Thus, we assumed that these species shared a similar preference for stop codon adoption and may have a closer relationship, as confirmed by the phylogenetic analysis (see below). The average AT contents of the 13 PCGs were 67.32% and 67.63% in *P. castaneus* and *P. laterotarsus*, respectively. The PCGs encoded by the J-strand displayed T-skews (T > A) and G-skews (G > C), whereas others encoded by the N-strand displayed T-skews and C-skews (C > G). The characteristics of the relative synonymous codon usages (RSCU) in these two new mitogenomes showed that codons including A or T at the third position were overrepresented compared with the other synonymous codons (Figure 2), reflecting the nucleotide bias of insect mitogenomes [8]. There was also a high AT content at the third codon site, indicating a high background mutational pressure towards AT nucleotides [60].

### 3.3. Evolutionary Rates of PCGs

Five available mitogenomes from the genus *Prosopocoilus* (from *P. castaneus*, *P. laterotarsus*, *P. confucius*, *P. astacoides*, and *P. blanchardi*) were used to calculate the evolutionary rate of its PCGs. For each PCG, the ratio of non-synonymous substitution (Ka) to synonymous substitution (Ks) was calculated (Figure 3). The NADH dehydrogenase subunits (*nad1*-*6* and *4l*) and ATP synthase subunits (*atp8* and *atp6*) had higher Ka/Ks ratios than the cytochrome oxidase subunits (*cox1*, *cox2**,* and *cox3*) and cytochrome b (*cytb*). This phenomenon suggested that various functional genes in the mitochondria of *Prosopocoilus* have undergone different selection pressures during the evolutionary process. The order of Ka/Ks for 13 protein genes was as follows: *atp8* > *nad6* > *nad5* > *nad4* > *nad4l* > *nad2* > *nad3* > *nad1* > *atp6* > *cox2* > *cytb* > *cox3* > *cox1*. Notably, the Ka/Ks ratios of the 13 PCGs among the five mitogenomes of *Prosopocoilus* were all less than 1.0, suggesting that these functional genes were all under strong purifying selection, as was reported in other insects [8,14,60]. The slowest and fastest evolution rates were observed for *cox1* and *atp8* genes, respectively (Figure 3). Furthermore, the *cox1* had the smallest evolutionary rate, indicating that positive selection was less powerful for this gene than functional constraints, as found in other insects or stag beetles [8,61,62].

### 3.4. Intergenic Spacers

In the two newly sequenced mitogenomes, the sizes of intergenic spacers were examined, varying from 1 bp to 375 bp in *P. castaneus*, and 1 bp to 158 bp in *P. laterotarsus* (Table 3). The large intergenic spacers in these *Prosopocoilus* mitogenomes were located between *trnI* and *trnQ*, consistent with those in the previously studied species of *P. blanchardi* (4051 bp), *P. astacoides* (375 bp), and *P. confucius* (580 bp) [4,23]. In Lucanidae, the notably large IGSs between *trnI* and *trnQ* were only found at this particular position in mitogenomes of the genus *Prosopocoilus* at present. Moreover, a short repetitive sequence with the sequence TAAAA was identified within the large IGS in both of the two newly sequenced mitogenomes. Compared with three previously reported *Prosopocoilus* species, the sites and number of times this short repetitive sequence (TAAAA) appeared and the lengths of the intergenic region when it was repeated were different among the five sequenced *Prosopocoilus* mitogenomes (Figure 4). In these five *Prosopocoilus* mitogenomes, the short sequence (TAAAA) appeared four times (*P. castaneus* and *P. laterotarsus*), five times (*P. blanchardi*), and three times (*P. confucius and P. astacoides*), and the length of the intergenic region between the repetitive sequence (TAAAA) differed among these five species (Figure 4). Additionally, in these five species of *Prosopocoilus*, we also detected an intergenic spacer of 18 bp between *trnS2*
*(UCN)* and *nad1* with the conserved motif (TACTAAA), similar to other reported lucanid beetles [4,20,50,51,52,53,54]. The IGS between *trnS2 (UCN)* and *nad1* is common in Coleoptera mitogenomes, but varies in length [4,20,23,50,51,52,53,54]. The characteristics of the large IGS between *trnI* and *trnQ* are only present in the representatives of this genus among the family Lucanidae; therefore, we propose that this feature may be synapomorphic for the members of the genus *Prosopocoilus*. This large IGS between *trnI* and *trnQ* could support the taxonomic positions of the other 195 species currently within *Prosopocoilus*. For example, *Epidorcus gracilis*, which was previously considered a species in the genus *Prosopocoilus*, does not contain this large IGS [4]. Previous studies have supposed that large intergenic spacers have possible roles in insect mitogenomic evolution owing to their existence in most insect species, despite their irregular appearance [55,56]. It is possible that this difference may have been caused by environmental selection pressures at the time during the evolutionary process and thus could serve as a useful phylogenetic signal [20]. The IGS between *trnI* and *trnQ* may be a useful marker for distinguishing *Prosopocoilus* from its closely related and indistinguishable genera with respect to this intergenic spacer that may exist in all studied *Prosopocoilus* species while it is absent in other genera. The present study has provided meaningful implications into the roles of these IGSs in the phylogenetic analysis of *Prosopocoilus* stag beetles. From our phylogeny is that the species that follow the genus *Prosopocoilus* have in common the presence of this large IGS that the implication could be in the role of this non-coding region in the evolution of the mitogenome of these species.

### 3.5. Phylogenetic Analyses

In this study, the nucleotide and amino acid sequences of the 13 PCGs were used to reconstruct the phylogenetic relationship through BI and ML inference methods and generate four nearly identical topologies (Figure 5 and Figure 6). Monophyly of the family Lucanidae was strongly supported (BPP = 1, MLP = 100), consistent with the phylogeny inferred from the multi-gene fragments in previous works [1,35]. Within the family Lucanidae (Figure 5 and Figure 6, blue shaded), close relationships were observed among *Prosopocoilus*, *Dorcus*, and *Serrognathus*. According to the topologies, the current genus *Prosopocoilus* was monophyletic and included the five representatives discussed in this study (BPP = 1, MLP = 100). *P. castaneus*, *P. astacoides*, and *P. blanchardi* formed the sister clade of the *P. confucius* and *P. laterotarsus* clade (Figure 5 and Figure 6, yellow shaded), whereas the subclade of *K. guangxi**i* was a sister group to the subclade of *E. gracilis* + *S. platymelus*. These two subclades were both included in *Dorcus* sensu lato. Additionally, from our phylogenomic tree, *Prosopocoilus* was a sister group to the clade ((*Hexarthrius* + *Rhaetus*) + *Pseudorhaetus*) consistent with the results of a study by Lin et al. [4]. The three species of the *D. velutinus* complex, *Dorcus curvidens hopei* and *Dorcus parapllelipipedus*, were clustered in a monophyletic clade, as reported by Chen et al. [20]. These results were also consistent with previous morphological comparisons [26,27]. Five species (*P. laterotarsus*, *P. castaneus*, *P. confucius*, *P. astacoides*, and *P. blanchardi*) shared high similarities with the type species of *Prosopocoilus* (*Prosopocoilus cavifrons* Hope & Westwood, 1845) in their external characteristics and the genitalia traits of both sexes [26,27], suggesting that these five representatives may follow the genus *Prosopocoilus* sensu stricto. The three others *K. guangxii*, *E. gracilis*, and *S. platymelus* showed partial morphological overlap with typical members of *Prosopocoilus*, *Dorcus,* and *Serrognathus*, and their taxonomic positions have been frequently adjusted among different genera by different taxonomists and amateurs [27,33,63,64,65]. Our phylogenetic analyses based on the mitogenomic data supported that the five species exhibited typical characteristics of *Prosopocoilus,* whereas the other three, which had partial morphological features, formed a different clade from the genus *Prosopocoilus*. We also found that *Dorcus* was a sister of the clade ((*Epidorcus* + *Serrognat**h**us*) + *Kirchnerius*). The relationships among these lucanid representatives have been described in previous studies [4,62]. The clade *Prosopocoilus* shares the presence of the large IGS, providing robust information for determining the taxonomic positions of the species following the genus *Prosopocoilus*. Additionally, our results continue to support that mitogenomes could be very useful molecular tools for improving our understanding of the phylogeny of problematic taxa in Lucanidae, despite the dense sampling materials needed.

## 4. Conclusions

Our study presents and describes the mitochondrial genomes of two stag beetles from the genus *Prosopocoilus*. The two mitogenomes of the genus *Prosopocoilus* maintained the typical gene content and organization of the ancestor mitogenome organization as proposed. The evolutionary rates of 13 PCGs among our studied *Prosopocoilus* (including three previously reported species) indicated that their evolution occurred under purifying selection. The large intergenic spacer was identified in each of the five *Prosopocoilus* mitogenomes, and comparisons suggested that the characteristics of large intergenic spacers (presence or absence, size, and location) may be phylogenetically meaningful for evaluating the genus *Prosopocoilus*. Our phylogenomic analyses including two newly sequenced species, further supported that *P. castaneus* and *P. laterotarsus* were clustered in a clade with typical *Prosopocoilus* species (*P. confucius*, *P. astacoides*, and *P. blanchardi*). Although we were unable to fully confirm that this large IGS was present in all *Prosopocoilus* species owing to limited sample availability, our findings established a new potential candidate for molecular identification of this genus. Moreover, our findings suggest that variations in the quantity, sequence, and location of IGSs may also be important signals for phylogenomic and evolutionary studies at lower taxonomic levels if these details become available for more taxa in the future. Finally, our results also indicated that mitogenomes could provide useful molecular evidence for improving our understanding of the evolution of stag beetles.

## Figures and Tables

**Figure 1 animals-12-01595-f001:**
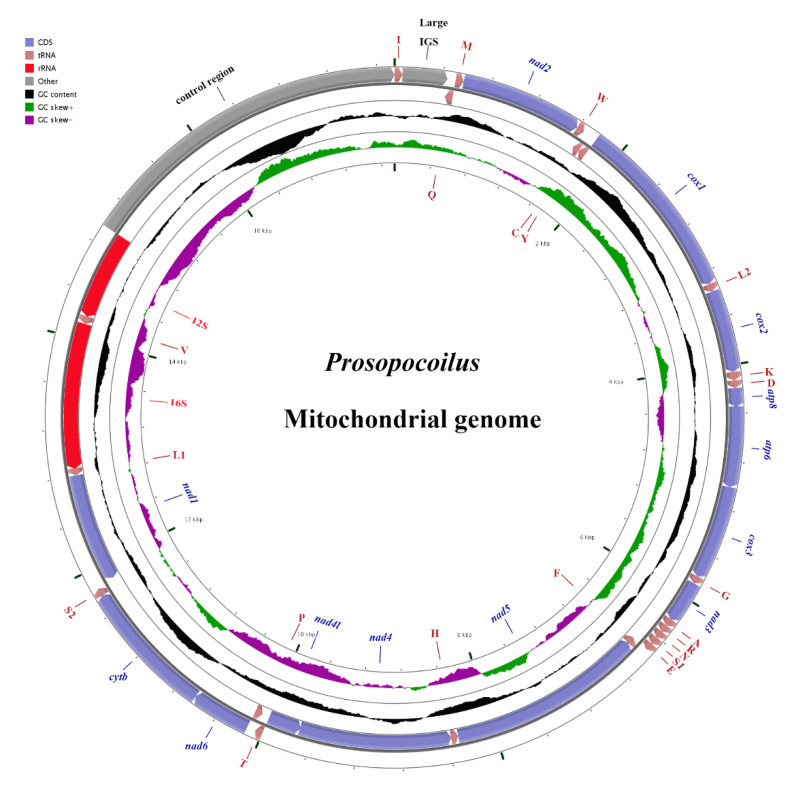
The mitochondrial genome map of *Prosopocoilus* species. The names of PCGs and rRNA genes are indicated using standard abbreviations, whereas the names of tRNA genes are indicated using single-letter abbreviations. The first circle shows the gene arrangement, and arrows indicate the direction of gene transcription. Blue, red, pink, and grey arrows indicate PCGs, rRNA genes, tRNA genes, and the control region, respectively. The second circle indicates the GC content, and the third circle indicates the GC-skew. The innermost circle indicates the size of the sequence.

**Figure 2 animals-12-01595-f002:**
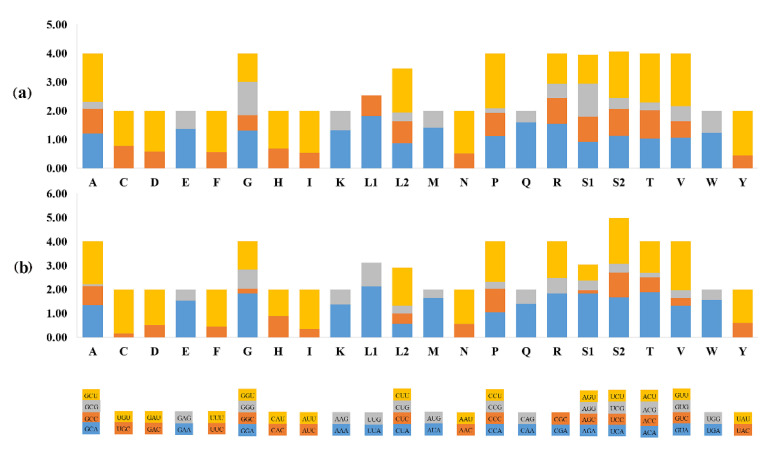
The relative synonymous codon usage (RSCU) in two newly sequenced species: (**a**) *P. castaneus*; (**b**) *P. laterotarsus*. The *X*-axis indicates the codons, and the *Y*-axis indicates the RSCU values.

**Figure 3 animals-12-01595-f003:**
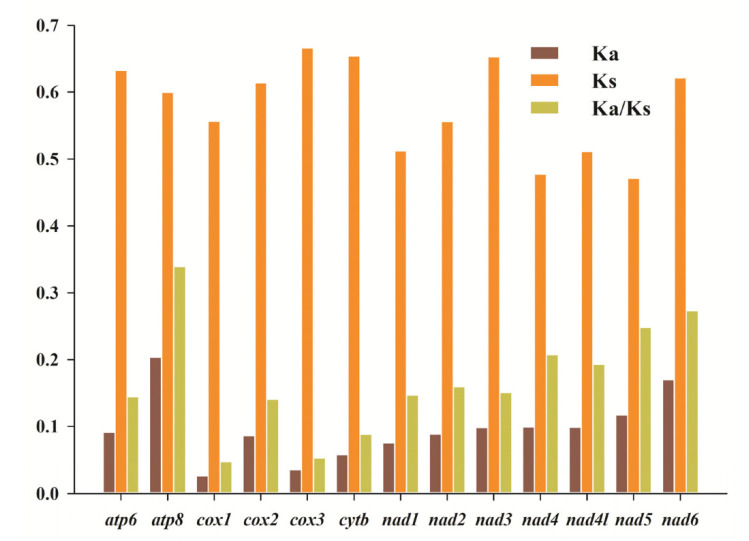
Evolution rate of mitochondrial protein-coding genes among five *Prosopocoilus* stag beetle species in the present study (*P. laterotarsus*, *P. castaneus*, *P. confucius*, *P. astacoides*, and *P. blanchardi*). Ka represents non-synonymous substitution rate, Ks represents synonymous substitution rate, and Ka/Ks represents the evolution rate of each PCG.

**Figure 4 animals-12-01595-f004:**
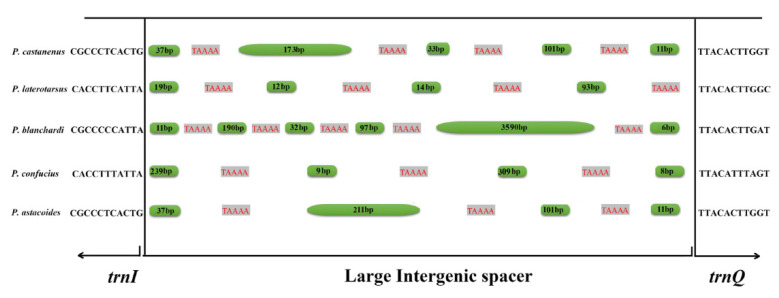
Composition of the large intergenic spacer between *trnI* and *trnQ* among the five sequenced mitochondrial genomes of *Prosopocoilus* in the present study. The grey-shaded region represents the short sequence repeat (TAAAA). The green-shaded region represents the size of the spacers between the short sequence (TAAAA) repeats.

**Figure 5 animals-12-01595-f005:**
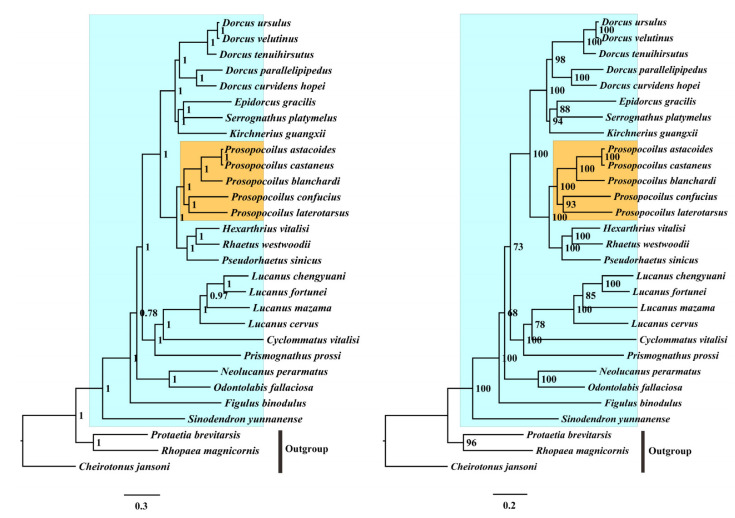
Phylogenomic relationships of Lucanidae inferred from mitogenomes. Three other scarab species (*Cheirotonus jansoni*, *Rhopaea magnicornis**,* and *Protaetia brevitarsis*) were used as the outgroup. The phylogenetic topology was obtained based on the data set PCGs. Blue shading shows Lucanidae and yellow shading shows *Prosopocoilus*. The numbers at the nodes are BI posterior probabilities (**left**) and ML bootstrap values (**right**).

**Figure 6 animals-12-01595-f006:**
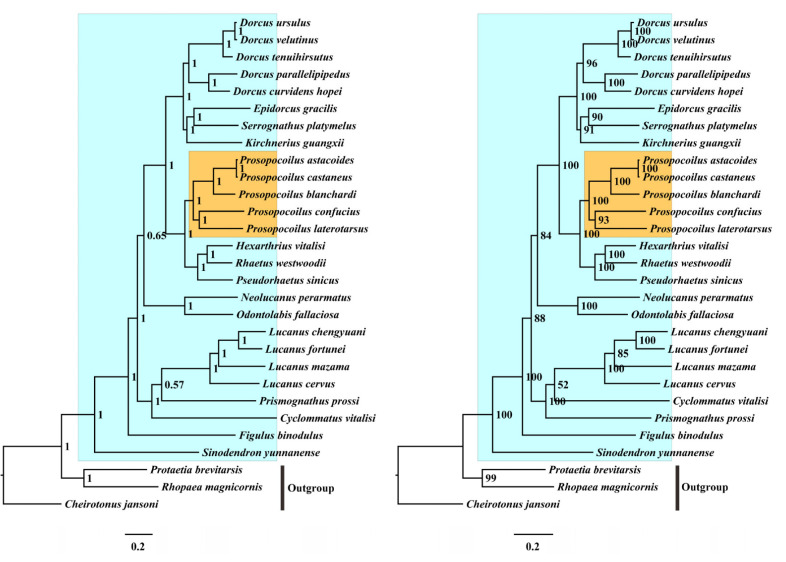
Phylogenomic relationships of Lucanidae inferred from mitogenomes. Three other scarab species (*Cheirotonus jansoni*, *Rhopaea magnicornis,* and *Protaetia brevitarsis*) were used as the outgroup. The phylogenetic topology was obtained based on the data set AA. Blue shading shows Lucanidae and yellow shading shows *Prosopocoilus*. The numbers at the nodes are BI posterior probabilities (**left**) and ML bootstrap values (**right**).

**Table 1 animals-12-01595-t001:** Primers used in this study.

Gene	Primer Name	Sequence (5′-3′)	Length (bp)	Reference
*cox1*	COI-F1	CAACATTTATTTTGATTTTTTGG	23	[4]
	COI-R1	TCCAATGCACTAATCTGCCATATTA	25	[4]
*cytb*	Cytb-F2	GAGGAGCAACTGTAATTACTAA	22	[4]
	Cytb-R2	AAAAGAAARTATCATTCAGGTTGAAT	26	[4]
*16S*	16S-F1	CCGGTTTGAACTCAGATCATG	21	[4]
	16S-R1	TAATTTATTGTACCTTGTGTATCAG	25	[4]

**Table 2 animals-12-01595-t002:** Accession numbers and lengths of mitochondrial genomes of species used in this study. I-Q indicates the presence of IGS between *trnI* and *trnQ*, whereas S2-1 indicates the presence of IGS between *trnS2 (UCN)* and *nad1*.

Family	Taxa	Length (bp)	I-Q/S2-1	ACC. Number	References
Lucanidae	*Cyclommatus vitalisi* (Pouillaude, 1913)	17,795	S2-1	MF037205	[20]
(Ingroup)	*Dorcus curvidens hopei* (Nomura, 1960)	16,026	S2-1	MF612067	[20]
	*Dorcus parallelipipedus* (Linnaeus, 1758)	17,563	S2-1	JX412841	[20]
	*Dorcus tenuihirsutus* (Kim & Kim, 2010)	18,266	S2-1	MK050991	[20]
	*Dorcus ursulus* (Arrow, 1938)	18,001	S2-1	MK050990	[20]
	*Dorcus velutinus* (Thomson, 1862)	16,939	S2-1	MK050989	[20]
	*Epidorcus gracilis* (Séguy, 1954)	16,736	S2-1	KP735805	[4]
	*Lucanus cervus* (Linnaeus, 1758)	20,109	S2-1	MN580549	[50]
	*Lucanus fortunei* (Saunders, 1854)	16,591	S2-1	JX313688	Unpublished
	*Lucanus mazama* (LeConte, 1861)	15,261	S2-1	FJ613419	[50]
	*Lucanus chengyuani* (Wang and Ko, 2018)	16,926	S2-1	MK878514	[51]
	*Figulus binodulus* (Waterhouse, 1873)	16,261	S2-1	NC045102	[52]
	*Kirchnerius guangxii* (Schenk, 2009)	15,296	S2-1	MK134567	[53]
	*Neolucanus perarmatus* (Didier, 1925)	16,610	S2-1	MF401425	Unpublished
	*Odontolabis fallaciosa* (Boileau, 1901)	20,276	S2-1	MF908524	[50]
	*Prismognathus prossi* (Bartolozzi & Wan, 2006)	15,984	S2-1	MF614014	[50]
	*Prosopocoilus blanchardi* (Parry, 1873)	21,628	I-Q/S2-1	KF364622	[23]
	*Prosopocoilus castaneus* (Hope et Westwood, 1845)	17,523	I-Q/S2-1	ON401054	This study
	*Prosopocoilus confucius* (Hope, 1842)	16,951	I-Q/S2-1	KU552119	[4]
	*Prosopocoilus laterotarsus* (Houlbert, 1915)	17,333	I-Q/S2-1	ON401055	This study
	*Prosopocoilus astacoides* (Hope, 1840)	17,746	I-Q/S2-1	NC050851	Unpublished
	*Pseudorhaetus sinicus* (Boileau, 1899)	18,126	S2-1	MZ504793	[54]
	*Hexarthrius vitalisi* (Didier, 1925)	18,362	S2-1	JX313676	Unpublished
	*Rhaetus westwoodi* (Parry, 1862)	18,131	S2-1	MG159815	[50]
	*Serrognathus platymelus* (Saunders, 1854)	17,088	S2-1	MF612070	Unpublished
	*Sinodendron yunnanense* (Král, 1994)	16,921	S2-1	KP735804	[4]
Scarabaeidae	*Cheirotonus jansoni* (Jordan, 1898)	17,249	S2-1	KC428100	[4]
(Outgroup)	*Protaetia brevitarsis* (Lewis, 1879)	20,319	S2-1	KC775706	[20]
	*Rhopaea magnicornis* (Blackburn, 1888)	17,522	S2-1	FJ859903	[50]

**Table 3 animals-12-01595-t003:** Mitogenomic organization of *P. castaneus* (*Pc*) and *P. laterotarsus* (*Pl*).

Gene	Strand	Region	Length (bp)	Start/Stop Codon	Intergenic (bp)
		Pc*/*Pl	Pc*/*Pl	Pc*/*Pl	Pc*/*Pl
*trnI*	J	1-64/335-398	64/64	-	375/158
*trnQ*	N	440-508/557-625	69/69	-	−1/−1
*trnM*	J	508-575/625-693	68/69	-	0/0
*nad2*	J	576-1589/694-1707	1014/1014	ATA(TAA)/ATA(TAG)	2/2
*trnW*	J	1592-1657/1710-1773	66/64	-	−8/−8
*trnC*	N	1650-1710/1766-1825	61/60	-	0/0
*trnY*	N	1711-1775/1826-1889	65/64	-	1/1
*cox1*	J	1777-3307/1891-3421	1531/1531	AAC(T)/AAT(T)	0/0
*trnL2 (UUR)*	J	3308-3372/3422-3485	65/64	-	0/0
*cox2*	J	3373-4060/3486-4170	688/685	ATA(T)/ATC(T)	0/0
*trnK*	J	4061-4130/4171-4240	70/70	-	0/0
*trnD*	J	4131-4193/4241-4303	63/63	-	0/0
*atp8*	J	4194-4349/4304-4459	156/156	ATT(TAA)/ATT(TAA)	−4/−4
*atp6*	J	4346-5014/4456-5124	669/669	ATA(TAA)/ATA(TAA)	−1/−1
*cox3*	J	5014-5797/5124-5907	784/784	ATG(T)/ATG(T)	0/0
*trnG*	J	5798-5861/5908-5970	64/63	-	0/0
*nad3*	J	5862-6213/5971-6322	352/352	ATG(T)/ATA(T)	0/0
*trnA*	J	6214-6278/6323-6387	65/65	-	−1/−1
*trnR*	J	6278-6341/6387-6450	64/64	-	−1/−1
*trnN*	J	6341-6404/6450-6513	64/64	-	0/0
*trnS1 (AGN)*	J	6405-6471/6514-6580	67/67	-	0/0
*trnE*	J	6472-6535/6581-6644	64/64	-	−2/−2
*trnF*	N	6534-6598/6643-6706	65/64	-	0/0
*nad5*	N	6599-8315/6707-8423	1717/1717	ATA(T)/ATT(T)	0/0
*trnH*	N	8316-8379/8424-8487	64/64	-	0/0
*nad4*	N	8380-9715/8488-9823	1336/1336	ATG(T)/ATG(T)	−7/−7
*nad4l*	N	9709-9996/9817-10104	288/288	ATG(TAA)/ATG(TAA)	2/2
*trnT*	J	9999-10061/10107-10171	63/65	-	0/−1
*trnP*	N	10062-10127/10171-10236	66/66	-	5/5
*nad6*	J	10133-10630/10242-10739	498/498	ATG(TAA)/ATG(TAA)	−1/−1
*cytb*	J	10630-11770/10739-11879	1141/1141	ATG(T)/ATG(T)	0/0
*trnS2 (UCN)*	J	11771-11835/11880-11944	65/65	-	18/18
*nad1*	N	11854-12804/11963-12913	951/951	ATA(TAG)/ATA(TAG)	0/0
*trnL1 (CUN)*	N	12805-12868/12914-12976	64/63	-	0/0
*rrnL*	N	12869-14136/12977-14242	1268/1266	-	0/0
*trnV*	N	14137-14205/14243-14311	69/69	-	−1/0
*rrnS*	N	14205-15014/14312-15116	810/805	-	0/−
Control region	-	15015-17523/-	2509/-	-	0/−

**Table 4 animals-12-01595-t004:** Base composition of two newly mitochondrial genomes.

Species	Genes	T(U)	C	A	G	A + T%	G + C%	AT-Skew	GC-Skew
*P. castaneus*	PCGs	39.14	17.06	28.19	15.61	67.32	32.68	−0.16	−0.04
	tRNAs	33.45	13.66	36.66	16.24	70.10	29.90	0.05	0.09
	rRNAs	39.03	8.86	31.42	20.70	70.44	29.56	−0.11	0.40
	Genome	32.07	20.33	36.67	10.93	68.73	31.27	0.07	−0.30
*P. laterotarsus*	PCGs	40.00	16.49	27.64	15.88	67.63	32.37	−0.18	−0.02
	tRNAs	34.20	12.31	37.55	15.94	71.75	28.25	0.05	0.13
	rRNAs	38.77	8.64	32.83	19.75	71.61	28.39	−0.08	0.39
	Genome	32.81	19.82	36.44	10.93	69.26	30.74	0.05	−0.29

## Data Availability

All the mitochondrial genome sequences were submitted to GenBank (accessions ON401054 and ON401055, as in Table 2), and they will be accessible when the article is published.

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
