# Peer review of "Phylogenetic Implication of Large Intergenic Spacers: Insights from a Mitogenomic Comparison of Prosopocoilus Stag Beetles (Coleoptera: Lucanidae)"

_animals, 2022, doi:10.3390/ani12131595_

Round 1
Reviewer 1 Report
Authors only report mitochondrial genome sequencing for Prosopocoilus. Reviewer thought the report should include the report of Prosopocoilus genome sequencing because we can sequence tens of giga bases of genome in a day. Reviewer cannot find any description about genome sequence of Prosopocoilus only sticking mitochondrial genome sequencing of two Prosopocoilus. Much more data including genome sequences must be required to discuss comparative analyses of insect species.
Author Response
Cover letter

Reviewer 2 Report
Dear authors,
here are my main comments on your valuable manuscript:
1) The quality of English language is pretty low throughout the manuscript, and specially in the simple summary, abstract and introduction. Some examples (not all) are given below:
L12-13: "...,to provide further in-12 sight into the relationships of its,..." - i really do not understand the meaning of this, and this should be the main goal of the manuscript
L15: "...all contain the large IGS that located..." - i assume that it should write "that is located" but i am not sure
There are more examples like this throughout the text.
L37-45: It is repeated multiple times in this paragraph that mitogenomes can be used for phylogeny. Please rewrite this section to be less redundant and more informative.
Figure 1 lacks description in the caption. What are all the symbols and bars representing?
Caption of the Figure 2 (first one) is not complete. There should be an explanation of the letters below the bars, and what is displayed on the y axis (in the label or in caption).
Figure 2 (second one) is labeled same as previous (so there are two Fig. 2 in the manuscript). This one also lacks the label (or caption) for y axis.
Figures 5-8 lack description (why different colors at some clades?). Also, do they really require full pages?
Author Response
Cover letter

Reviewer 3 Report
Dear authors,I appreciate the effort you have made, it is an interesting study that can be improved. This manuscript can be considered to this journal after various modification. I did in the pdf file various questions, comments, recommendations and modifications. You need to improve the organization of tables, and more important improved the phylogenetic results.
You should check in detail the annotation of the mitogenomes you sequenced. I am very surprised that you have not found the IGS between S2 and nad 1, this is a common IGS in Coleoptera, check more literature. I have pointed out to you that another species within this genus (Prosopocoilus confucius) which has this IGS. Has this been verified by you on all the lucanids included ?
You should improve the phylogenetic results, there is really confusion of concepts and terminology and it seems to me the weakest section of your work. You should include the evolutionary models that were suggested by partitionfinder. The matrix of concatenated DNA sequences should be available in some database. This is useful for readers to repeat and check their results. I really have doubts with this statement in the title, because your conclusion indicate that the IGS could be used in phylogeny, but you don't do phylogeny using this mark. Actually your result shown that the species which are grouped in the genus (4 / 200 species), and which probably follow the genus sensu stricto (I don't know which is the type species and is already included in a phylogeny) share in common a long IGS between tRNAs I and Q; now, the evolutionary implication of this IGS is little discussed.

Author Response
Cover letter

Round 2
Reviewer 1 Report
Thanks for your explanation. Reviewer understands the situation in your target insects. Be sure to open the data you submitted to GenBank after the publication.
Reviewer 2 Report
Dear Authors,
thank you for revising your manuscript and adding additional information where necessary. Readability of the manuscript is now also highly improved and i believe it can be accepted for publishing in its current form.
Best wishes